# Dietary Inulin Supplementation Affects Specific Plasmalogen Species in the Brain

**DOI:** 10.3390/nu14153097

**Published:** 2022-07-28

**Authors:** Jean-Baptiste Bizeau, Mayssa Albouery, Stéphane Grégoire, Bénédicte Buteau, Lucy Martine, Marine Crépin, Alain M. Bron, Olivier Berdeaux, Niyazi Acar, Benoit Chassaing, Marie-Agnès Bringer

**Affiliations:** 1Eye and Nutrition Research Group, Centre des Sciences du Goût et de l′Alimentation, Institut Agro Dijon, CNRS, INRAE, Université Bourgogne Franche-Comté, F-21000 Dijon, France; jean-baptiste.bizeau@inrae.fr (J.-B.B.); mayssabouery@gmail.com (M.A.); stephane.gregoire@inrae.fr (S.G.); benedicte.buteau@inrae.fr (B.B.); lucy.martine@inrae.fr (L.M.); alain.bron@chu-dijon.fr (A.M.B.); niyazi.acar@inrae.fr (N.A.); 2ChemoSens Platform, Centre des Sciences du Goût et de l′Alimentation, Institut Agro Dijon, CNRS, INRAE, Université Bourgogne Franche-Comté, F-21000 Dijon, France; marine.crepin@inrae.fr (M.C.); olivier.berdeaux@inrae.fr (O.B.); 3Department of Ophthalmology, University Hospital, Dijon, F-21000 Dijon, France; 4Inserm U1016, Team “Mucosal Microbiota in Chronic Inflammatory Diseases”, CNRS UMR 8104, Université Paris Cité, 75014 Paris, France; benoit.chassaing@inserm.fr

**Keywords:** dietary fibers, inulin, lipid, glycerophospholipid, plasmalogen, fatty acid, docosahexaenoic acid, brain, cortex, liver

## Abstract

Plasmalogens (Pls) are glycerophospholipids that play critical roles in the brain. Evidence supports the role of diet and that of the gut microbiota in regulating brain lipids. We investigated the impact of dietary intake of inulin—a soluble fiber used as prebiotic—on the Pl content of the cortex in mice. No global modification in the Pl amounts was observed when evaluated by gas chromatographic analysis of dimethyl acetals (DMAs). However, the analysis of individual molecular species of Pls by liquid chromatography revealed a reduced abundance of major species of ethanolamine Pls (PlsEtn)―PE(P-18:0/22:6) and PE(P-34:1)―in the cortex of mice fed a diet supplemented with inulin. DMA and expression levels of genes (*Far-1*, *Gnpat, Agps*, *Pla2g6* and *Tmem86b*) encoding key enzymes of Pl biosynthesis or degradation were not altered in the liver and in the cortex of mice exposed to inulin. In addition, the fatty acid profile and the amount of lyso forms derived from PlsEtn were not modified in the cortex by inulin consumption. To conclude, inulin affects the brain levels of major PlsEtn and further investigation is needed to determine the exact molecular mechanisms involved.

## 1. Introduction

The brain is the second-richest organ in terms of lipid content after adipose tissue. Lipids account for about half of the dry weight of the brain and are essential components in the structure and function of this organ [1,2,3]. The crucial role of lipids in maintaining the health status of the brain is well illustrated by the existence of neurological disorders (e.g., mood disorder, bipolar disorders and schizophrenia) and neurodegenerative diseases (e.g., Alzheimer’s disease (AD) and Parkinson’s disease (PD)) that are associated with alterations in lipid homeostasis in the brain [1,2]. In addition to displaying a high lipid content, the brain is also characterized by a high lipid diversity, which relies mainly on fatty acids [4,5]. In the brain, phospholipids are the reservoirs of fatty acids, and particularly of arachidonic acid (ARA, C20:4*n*-6) and docosahexaenoic acid (DHA, C22:6*n*-3), which are polyunsaturated fatty acids (PUFAs) involved in the regulation of the structure and functions of brain cells [6,7]. Indeed, in addition to serving as an energy source, fatty acids also act as structural components of the cell membrane, and their derivatives are involved in cell signaling processes [6].

One of the specific features of brain phospholipid composition is its enrichment in a unique subclass of glycerophospholipids termed “plasmalogens” (Pls) [8]. Pls are characterized by a vinyl ether bond linking a long-chain fatty alcohol to the glycerol backbone in the *sn*-1 position of glycerol instead of an ester bond as found in other glycerophospholipids. The fatty alcohols in Pls are C16:0, C18:0 and C18:1 (*n*-7 or *n*-9). The fatty acid esterified at the *sn*-2 position of glycerol is predominantly ARA or DHA. In the brain, the polar head at the *sn*-3 position of the glycerol molecule is mainly ethanolamine. The brain has the highest content of ethanolamine Pls (or plasmenyl-ethanolamine, PlsEtn), which account for more than half of the total ethanolamine phospholipids in this tissue [8]. Pls can be derived from dietary intake and/or can be endogenously synthesized in tissues [9,10,11]. Moreover, some studies suggest that the liver might provide Pls for other tissues, but this concept remains controversial. Indeed, the amount of Pls and the level of activity of enzymes involved in their biosynthesis were found to be very low in the liver [8,12]. Pls are carried into the blood via chaperone proteins, low-density lipoprotein (LDL) being the major carrier. Pls are thereafter delivered to tissues through the LDL receptor pathway [13]. Studies suggest that Pls and their precursors might cross the blood–brain barrier, but incorporation of dietary Pls or their precursors in the brain does not seem to be as efficient when compared to peripheral organs [14,15,16]. The endogenous synthesis of Pls in the brain is thought to be the main source of brain Pls but this is still under debate [12,17]. The biosynthesis of Pls starts in peroxisomes and ends in the endoplasmic reticulum (Figure 1). Critical steps in Pl biosynthesis include the reduction of fatty acid to fatty alcohol by fatty acyl-CoA reductase 1 (encoded by *Far1*), which is an enzyme located in the outer surface of the peroxisomal membrane, and by the two peroxisomal enzymes DHAP-AT/DAP-AT (dihydroxyacetone phosphate acyltransferase, encoded by *Gnpat*) and alkyl-DHAP synthase (alkylglycerone-phosphate synthase, encoded by *Agps*) [12].

Several functions and properties have been attributed to Pls. The high susceptibility of the vinyl ether bond to oxidative damage has been described as a property of Pls that may protect other lipids in cell membranes and lipoproteins against oxidative stress. However, this hypothesis is controversial [18]. Pls also influence the physical and chemical properties of biomembranes (e.g., fluidity, thickness and lateral pressure) and thereby cellular and subcellular processes such as vesicle formation and membrane fusion events [17]. The enrichment of “lipid rafts” with Pls may also affect the initiation of signal transduction in membranes [19]. Pls are involved in the composition of glycosyl-phosphatidyl-inositol anchors, a post-translational modification of membrane proteins [20]. In addition, they constitute reservoirs of biologically active lipid mediators that are produced subsequently to the release of ARA and DHA by phospholipase A2 hydrolysis [21,22]. Lysoplasmalogens that are generated following the fatty acyl cleavage from Pls may also have biological functions, both as precursors of Pls and as metabolites [21]. Pls have been identified as a major structural component of the brain, and particularly of myelin and synaptic membranes. They also modulate processes that are important for maintaining brain homeostasis and functions such as neurotransmission, oxidative stress and neuroinflammation. The important role of Pls in brain physiology is highlighted by the association of neurological diseases/disorders with an abnormal composition or with abnormal levels of Pls or of enzymes involved in their biosynthesis [16]. In addition, beneficial effects of supplementation with Pls/Pl precursors on brain functions have been reported, particularly in the context of AD [11,23].

It is well documented that the gut microbiota influences the physiology of organs at distance from the gut mucosa, including the nervous tissues [24,25]. In particular, there is evidence that the gut microbiota modulates the lipid composition of both the brain and the retina—the neurosensorial tissue that lines the back of the eye and that is known to be an extension of the central nervous system. Indeed, analysis of the retinal lipidome of germ-free mice and conventionally raised mice showed that the gut microbiota influences the PlsEtn content of the retina [26]. In addition, comparison of the lipid profile of germ-free mice colonized with the gut microbiota of young or old donor mice revealed that the composition of the gut microbiota affects the cholesterol and phospholipid content of the cortex, including phosphatidylcholine (PChol), phosphatidylethanolamine (PEtn) and PlsEtn species [27].

Diet and the gut microbiota are intrinsically linked [28]. Among dietary factors shaping the gut microbiota and influencing its functions is the consumption of dietary fibers [29]. Dietary fibers can be categorized according to their water solubility. Whereas insoluble fibers (e.g., cellulose or hemicellulose) are poorly digested in the colon by the gut microbiota, soluble fibers (e.g., inulin-type fructans) can be fermented by gut bacteria. The fermentation of soluble fibers by bacteria generates metabolites (e.g., short-chain fatty acids (SCFAs)) that can have biological effects on the host, including effects on lipid metabolism [30]. A lack of fibers has been shown to alter the composition, diversity and richness of the gut microbiota [31,32,33]. Soluble dietary fibers may influence the gut microbial ecosystem in several ways. The consumption of soluble dietary fibers favors not only the expansion of gut bacteria that are enzymatically equipped to degrade these substrates, but also that of gut bacteria that will take advantage of the physicochemical changes associated with the presence of fibers (e.g., acid environment) and/or benefit from the intermediate products or metabolites arising from the fiber degradation. The influence of inulin on the gut microbiota is particularly well documented. Data obtained from mouse models as well as from studies of humans showed that inulin consumption is associated with the expansion of bacteria that are described as conferring health benefits and with a reduction in pathobionts [34,35,36,37]. Modulation of the host lipid metabolism is also associated with inulin consumption. Indeed, effects of inulin on triglyceride and cholesterol blood levels have been reported, but these findings are still controversial [38,39,40]. In addition, we recently showed that supplementation of a low- or high-fat diet with inulin affects the fatty acid content of mouse liver [34]. Although no direct causal relationship has been established, some inulin-induced changes in the gut microbiota were correlated with modification of the expression of genes encoding enzymes involved in fatty acid biosynthesis [34]. The aim of this study was to investigate whether dietary intake of inulin affects the Pl content of the brain. To this end, mice were exposed to a diet supplemented with either cellulose or inulin. The abundance and the diversity of Pls were explored in the liver and the cortex of mice through gas and liquid chromatographic techniques. The expression levels of the key enzymes involved in Pl biosynthesis and cleavage/degradation were also determined.

## 2. Materials and Methods

### 2.1. Mice and Diets

For this study, 5-week-old male C57BL/6J mice were purchased from The Jackson Laboratory (Bar Harbor, ME, USA). Mice were housed at Georgia State University, Atlanta, GA, USA until euthanasia under institutionally approved protocols (Institutional Animal Care and Use Committee IACUC #A18006). Mice were maintained on 12 h light:dark cycles with ad libitum access to food and water. After 1 week of acclimation, mice were randomly divided into two groups: a control group (CTRL; *n* = 12) received a purified diet supplemented with 50 g cellulose/kg (Research Diet; #D12450J) and an inulin group (INU; *n* = 11) received a purified diet supplemented with 200 g inulin/kg (Research Diet; #D13081108) [34]. Cellulose as a source of fiber is generally poorly fermented by the gut. The diet containing cellulose served as a control. The source of inulin was chicory (average degree of polymerization ≥ 23; Orafti^®^ HP; BENEO-Orafti, Tienen, Belgium). Mice were maintained on these respective diets for 11 weeks. Blood was collected by retrobulbar venous plexus puncture in heparinized tubes and plasma was isolated after centrifugation (1800× *g*, 10 min, 4 °C). They were then euthanized by cervical dislocation and the cortex and liver were collected.

### 2.2. Lipid Extraction and Determination of Fatty Methyl Ester and Dimetyl Acetal Profiles

Total lipids from cortex, plasma and livers were extracted using Folch’s procedure [41]. Boron trifluoride in methanol was used for transmethylation [42]. Hexane was used to extract fatty acid methyl esters (FAMEs) and dimethyl acetals (DMAs). Analyses were performed on a GC Trace 1310 (Thermo Scientific, Illkirch, France) gas chromatograph (GC) using a CPSIL-88 column (100 m × 0.25 mm inside diameter, film thickness 0.20 μm; Agilent, CA, USA). This device was coupled to a flame ionization detector (FID). The configuration was: inlet pressure of hydrogen 210 kPa, oven temperature 60 °C for 5 min + 165 °C at 15 °C per min and upholding for 1 min, +225 °C at 2 °C per min and upholding at 225 °C for 17 min. The injector and the detector were maintained at 250 °C. Comparisons with commercial and synthetic standards enabled the identification of FAMEs and DMAs. The ChromQuest 5.0 version 3.2.1 software (Thermo Scientific, Illkirch, France) was used to process the data.

### 2.3. Analysis of Phospholipid Molecular Species

The phosphorus content of the total lipid extract was determined according to the method developed by Bartlett and Lewis [43]. The total phospholipids were dried under a stream of nitrogen and diluted to the appropriate concentration of 500 µg/µL of phospholipids in chloroform/methanol (CHCl_3_/CH_3_OH) (1:1, *v*/*v*). Ten microliters of internal standard mixture containing PC(14:0/14:0) 320 µg/mL, PE(14:0/14:0) 160 µg/mL, PS(14:0/14:0) 80 µg/mL, PI(8:0/8:0) 100 µg/mL and SM(d18:1/12:0) 80 µg/mL were added into 200 µL of this phospholipid solution.

The process of identification and quantification of phospholipid species was performed on a Thermo UltiMate™ 3000 coupled to an Orbitrap Fusion^TM^ Tribrid Mass Spectrometer equipped with an EASY-MAX NGTM Ion Source (H-ESI) (Thermo Scientific, Waltham, MA, USA).

Separation of phospholipid classes was achieved under hydrophilic interaction liquid chromatography (HILIC) conditions using a Kinetex HILIC 100 m × 2.1 mm, 1.7 µm column (Phenomenex, Sydney, Australia), with a flow of 0.5 mL/min. The mobile phase consisted of (A) acetonitrile/water (CH_3_CN/H_2_O) (96:4, *v*/*v*) containing 10 mM ammonium acetate and (B) CH_3_CN/H_2_O (50:50, *v*/*v*) containing 10 mM ammonium acetate. The chosen solvent-gradient system of the analytical pump was as follows: 0 min 100% A, 12 min 80% A, 18 min 50% A, 18.1–30 min 100% A. The injection volume was 10 µL and the column was maintained at 50 °C.

Phospholipid species were detected by high-resolution mass spectrometry (HRMS) analysis. H-ESI source parameters were optimized and set as follows: ion transfer tube temperature of 285 °C, vaporizer temperature of 370 °C, sheath gas flow rate of 35 au, sweep gas of 1 au, auxiliary gas flow rate of 25 au. Positive and negative ions were monitored alternatively by switching the polarity approach with a static spray voltage at 3500 V and 2800 V in positive and negative mode, respectively. Mass spectra in full scan mode were obtained using the Orbitrap mass analyzer with the normal mass range and a target resolution of 240,000 (full width at half maximum (FWHM) at *m*/*z* 200), in a mass-to-charge ratio *m*/*z* ranging from 200 to 1600 using a Quadrupole isolation in a normal mass range. All mass spectrometry (MS) data were recorded using a maximum injection time of 100 ms, automatic gain control (AGC) target (%) at 112.5, radio frequency lens (%) at 50 and one microscan. An intensity threshold filter of 1.103 counts was applied.

For tandem mass spectrometry (MS/MS) analyses, the data-dependent mode was used for the characterization of phospholipid species. Precursor isolation was performed in the Quadrupole analyzer with an isolation width of *m*/*z* 1.6. Higher-energy collisional dissociation was employed for the fragmentation of phospholipid species with an optimized stepped collision energy of 27%. The linear ion trap was used to acquire spectra for fragment ions in data-dependent mode. The AGC target was set to 2.104 with a maximum injection time of 50 ms. All MS and MS/MS data were acquired in the profile mode.

The Orbitrap Fusion was controlled by Xcalibur^TM^ 4.1 software (Thermo Scientific, Waltham, MA, USA). Data of high accuracy and the information collected from fragmentation spectra, with the help of the LipidSearch^TM^ 2.0 software (Thermo Scientific, Waltham, MA, USA) and the LIPID MAPS^®^ database [44], were used for phospholipid species identification.

### 2.4. Gene Expression

Total RNA was extracted using TRIzol reagent (Fisher Scientific, Illkirch, France). Reverse transcription was performed with the PrimeScript RT reagent kit containing gDNA Eraser (Takara Bio Europe, Saint Germain-En-Laye, France) and using 500 ng of total RNA. Gene expression was determined by real-time polymerase chain reaction (PCR) using SYBR Green (Bio-Rad, Marnes-La-Coquette, France) and a CFX96 Real-Time PCR system (Bio-Rad, Marnes-La-Coquette, France). *Hprt* was used as the internal control for normalization. Fold induction was calculated with the delta-delta Ct (ddCt) method. Primer sequences are given in Table 1.

### 2.5. Statistical Analysis

Statistical analyses were performed using Prism 6 software (GraphPad Software Inc., San Diego, CA, USA). The non-parametric Mann–Whitney test was used to compare data from the two groups. All *p* values of less than 0.05 were considered statistically significant (* *p* < 0.05, ** *p* < 0.01, *** *p* < 0.001, and **** *p* < 0.0001).

## 3. Results

### 3.1. Effect of Inulin on the Level of Total Pls in the Liver, in the Plasma and in the Cortex

The liver has been proposed as the primary organ of Pl biosynthesis. However, in contrast to the brain whose Pl content is very high, the hepatic level of Pls is very low due to a low storage rate and a high rate of export to other organs [8,12]. The amount of Pls in the liver and in the cortex was measured by GC-FID. *Acid-catalyzed* transmethylation of the aldehyde aliphatic groups from the *sn*-1 position of Pls resulted in the production of DMAs (DMA 16:0, DMA 18:0, DMA 18:1*n*-7 and DMA 18:1*n*-9) whose amounts could be determined concomitantly with FAMEs by GC-FID. As expected, we observed that the amount of DMAs in the liver of control mice represented only 0.06% ± 0.005% of the total FAMEs and DMAs (Figure 2a). Only one class of DMAs was detected: DMA 16:0 (Figure 2a). Supplementation of the diet with inulin did not modify the hepatic level of DMA 16:0 (Figure 2a).

The level of total Pls was also measured in the plasma. As for the liver, the mean level of DMAs in the plasma was low (0.80% ± 0.10% of total DMAs and FAMEs in CTRL mice; Figure 2b). Inulin did not modify the total amount of DMAs in this transport fluid (Figure 2b). However, the analysis at the species level revealed that the relative abundance of the two DMA species detected in the plasma (DMA 16:0 and DMA 18:0) was modified by inulin consumption: the plasma level of DMA 16:0 was significantly decreased and that of DMA 18:0 significantly increased in the plasma of INU mice compared to CTRL mice (Figure 2c,d).

In the cortex, the amount of total DMAs represented 9.24% ± 0.11% of the total FAMEs and DMAs (Figure 2e). Four DMA classes were detected (Figure 2f–i). Among DMAs, DMA 18:0 was the most widely represented (44.08% ± 0.51% of total DMAs in CTRL mice, Figure 2g), followed by DMA 16:0 (23.34% ± 0.26%, Figure 2f), DMA 18:1*n*-7 (17.02% ± 0.31%, Figure 2h), and DMA 18:1*n*-9 (15.55% ± 0.18%, Figure 2i). No effect of inulin was observed neither on the amount of total DMAs in the cortex nor on the amounts of individual subclasses of DMAs (Figure 2e–i).

These data indicate that, despite its effects on Pl classes in the plasma, inulin had no impact on the total amount of Pls in the cortex.

### 3.2. Impact of Dietary Supplementation with Inulin on the Plasmalogen Content of the Cortex

#### 3.2.1. Overview of Plasmalogen Species

In the cortex, PlsEtn are the most abundant Pls [8,45]. A total of 102 glycerophospholipid species were identified in the cortex of control mice by liquid chromatography-tandem mass spectrometry method (HPLC-MS^2^) analyses. Among them, five were alkyl-glycerophospholipids (AKGs), which are intermediate molecules in the biosynthesis of Pls, and 16 were alkenyl-glycerophospholipids, namely, Pls (Figure 1 and Table 2). As expected, the large majority (76.2%) of AKGs and Pls belonged to the ethanolamine subclass (Table 2). PlsEtn represented 46.155 ± 1.303% of the overall ethanolamine glycerophospholipid species. The three most abundant PlsEtn were PE(P-18:0/22:6), PE(P-16:0/22:6), and PE(P-18:0/20:4), which represented 10.857 ± 0.530%, 5.189 ± 0.299%, and 5.001 ± 0.230% of total ethanolamine glycerophospholipids in CTRL mice, respectively (Table 2).

Among the other glycerophospholipids, we identified three AKG species in the choline (AKGChol) and one in the inositol subclasses (Table 2). Only one Pl species, PC(P-32:0), was detected in the class of choline glycephospholipids. This represented only 0.125% ± 0.010% of total choline glycerophospholipids (Table 2).

#### 3.2.2. Impact of Dietary Inulin Supplementation on the Abundance of Plasmalogen Species in the Liver and the Cortex

For each species and each glycerophospholipid class, we compared the abundance of the individual species of AKGs and Pls in the cortex of INU mice relative to that of CTRL mice (Figure 3).

Whereas the abundance of the AKGEtn (AKG species in the ethanolamine subclass) and AKGChol was unchanged, an 18.5% ± 6.0% increase in the abundance of PI(O-16:0/20:4) was observed in the cortex of INU mice compared with CTRL mice (Figure 3a). In addition, among the 15 PlsEtn species identified, two were significantly decreased in the cortex of INU mice compared with CTRL mice (Figure 3b). Indeed, inulin supplementation was associated with a 20.5% ± 5.5% decrease in PE(P-18:0/22:6), which is the most abundant PlsEtn species in the cortex, and with a 15.7% ± 6.8% decrease in PE(P-34:1) [PE(P-16:0/18:1); PE(P-18:1/16:0)] (Figure 3b). The abundance of PlsChol was not modified by inulin (Figure 3b).

Altogether, these results show that supplementation of the diet with inulin modifies the abundance of specific AKG and individual Pl species in the cortex.

### 3.3. Effect of Inulin on the Expression of Genes Encoding Enzymes Involved in Plasmalogen Biosynthesis

Fatty acyl-CoA reductase 1 (encoded by *Far1*), alkyl-DHAP synthase (encoded by *Agps*), and DHAP-AT/DAP-AT (encoded by *Gnpat*) are key enzymes involved in Pl biosynthesis (Figure 1). As their level of expression could be a factor modulating the amount of Pl, we compared the mRNA levels encoding these enzymes in the liver (Figure 4a) and in the cortex (Figure 4b) of INU and CTRL mice. As estimated by the comparison of the DeltaCt (ΔCt), the expression levels of *Far1*, *Agps* and *Gnpat* in CTRL mice were significantly lower in the liver than in the cortex (Appendix A Figure A1). Diet supplementation with inulin did not modulate gene expression in either organ (Appendix A Figure A1 and Figure 4).

### 3.4. Modulation of the Fatty Acid Content of the Cortex by the Dietary Intake of Inulin

Another limiting factor that could have affected the amounts of PlsEtn PE(P-18:0/22:6) and PE(P-34:1) [PE(P-16:0/18:1); PE(P-18:1/16:0)] in the cortex is the bioavailability of fatty acids entering the biosynthesis of these lipid species. Therefore, we analyzed the fatty acid composition of the cortex by GC-FID in INU mice compared with CTRL mice (Table 3). We observed that inulin supplementation has a weak effect on the saturated fatty acid (SFA) content of the cortex, since only the abundance of two minor SFAs (C15:0 and C17:0) was significantly modified (Table 3). Among monounsaturated fatty acids (MUFAs), a trend toward a decrease in the amount of total MUFAs of the *n*-7 series (*p* = 0.0572) and a significant decrease in the abundance of C16:1*n*-7 were observed in the cortex of mice exposed to inulin compared to those fed a control diet (Table 3). However, the dietary intake of inulin modulated the abundance of several PUFAs in the cortex. The abundance of C22:5*n*-3, C18:2*n*-6 and C20:3*n*-6 was decreased whereas that of C22:5*n*-6 and C20:3*n*-9 was increased in the cortex of INU mice compared to CTRL mice (Table 3).

These results indicate that feeding mice an inulin-supplemented diet influences the fatty acid content of the cortex. However, alterations induced by inulin did not concern the fatty acids involved in the composition of PlsEtn PE(P-18:0/22:6) and PE(P-34:1) [PE(P-16:0/18:1); PE(P-18:1/16:0], namely, C16:0, C18:0, C18:1 and C22:6*n*-3.

### 3.5. Influence of Inulin on the Production of Lyso-Glycerophospholipids in the Cortex

A decrease in the amounts of glycerophospholipids can result from an enhanced production of metabolic intermediates termed “lyso-glycerophospholipids” that are generated by the release of the fatty acid esterified at the *sn*-2 position of the glycerol molecule following the action of the enzyme phospholipase A(2) encoded by the *Pla2g6* gene [46]. The vinyl-ether bond of lysoplasmalogens can then be cleaved by the enzyme lysoplasmalogenase encoded by the *Tmem86b* gene. As estimated by the analysis of the ΔCt levels, the expression levels of these genes was significantly higher in the liver than in the cortex (Appendix A Figure A1). We observed no modification of the expression levels of *Pla2g6* and *Tmem86b* in liver and cortex of INU mice compared to CTRL mice (Figure 5).

Using HPLC-MS^2^, we analyzed and compared the amounts of lyso-ethanolamine glycerophospholipids in the cortex of mice fed control or inulin-supplemented diet (Table 4). In total, 21 species of lyso-phosphatidylethanolamine (LPEs) species were identified but no lyso form of PlsEtn was detected (Table 4). No significant modification of the ratio of total LPEs/total ethanolamine glycerophospholipids was observed in the cortex of CTRL mice compared to that of INU mice (Table 4). In addition, we observed at the individual species level that the INU diet affected the abundance of LPE 14:0 (Table 4).

Altogether, these results suggest that dietary intake of inulin is not associated with an increase in LPEs in the cortex.

### 3.6. Influence of Inulin on Oxidative Stress-Related Mechanisms in the Cortex

As oxidative-stress-related molecules could cause Pl degradation by attacking their vinyl-ether bond [47,48], we compared the expression level of a set of genes involved in oxidative stress-related mechanisms: *Cat* encoding catalase, *Gpx1* encoding for glutathione peroxidase 1, *Nos2* encoding for inducible NO synthase, *Sod1* encoding for superoxide dismutase (Cu-Zn), *Cox-2* encoding for cyclooxygenase-2 and *Sqstm1* encoding for sequestosome-1 (ubiquitin-binding protein p62). As presented in Figure 6, we did not observe any modification in the expression levels of these genes in the cortex of mice from the INU group compared to CTRL mice, suggesting that oxidative stress-related mechanisms were not modulated by inulin.

## 4. Discussion

The brain is highly enriched in Pls, where they are essential in maintaining structure (e.g., myelination) and homeostasis (e.g., anti-oxidative properties, regulation of inflammation) as well as the functioning of specific processes (e.g., neurotransmission) [16]. It is now well recognized that the lipid composition of the brain is modulated by the lipid composition of the diet [49,50,51,52,53]. Thanks to the efforts made over the past decades to understand the link between diet, gut microbiota, and host metabolism, it has become evident that diet modulates not only host lipids by bringing lipids and their precursors to the host but also by acting through the gut microbiota [54]. A body of evidence indicates that the gut microbiota influences different aspects of host lipid metabolism as well as the lipid composition of organs, including that of the brain [27,55,56,57,58]. In this study, we investigated the effect of inulin, a soluble dietary fiber with prebiotic properties, on the content and composition of Pls in the brain.

Determination of the DMA profile in the cortex of mice fed a control diet revealed that Pls represent approximately 9.3% of total fatty acids in this brain structure and that they are distributed into four classes (DMA 16:0, DMA 18:0, DMA 18:1*n*-7 and DMA 18:1*n*-9), with DMA 18:0 and DMA 16:0 being the most abundant. These results are in agreement with previous studies [59]. We found that the dietary supplementation with inulin did not alter the DMA content or the distribution of DMAs into the different classes, suggesting that dietary intake of this prebiotic does not affect the whole Pl content of the cortex, or the distribution of Pls according to their *sn*-1 position.

In addition to the fatty alcohol moiety linked by a vinyl–ether bond at the *sn*-1 position (whose trans-methylation yields the DMA derivatives), the diversity of Pl species is also ensured by the fatty acid esterified at the *sn*-2 position as well as by the polar head group at the *sn*-3 position of glycerol. Analysis by liquid chromatography coupled to MS/MS of the diversity of glycerophospholipids in the cortex of mice enabled the identification of five AKG species that are intermediate metabolites of Pl synthesis, as well as 16 species of Pls. As reported in other studies, we observed that most of them (76.2%) were PlsEtn [8]. It has been shown in the context of PlsEtn deficiency that the level of PEtn is adjusted to keep the level of PlsEtn + PEtn constant [60]. In our study, no modification in the total amount of PlsEtn or PEtn was observed in the cortex of mice fed an inulin-supplemented diet compared to those fed a control diet. However, two Pl species were affected by supplementation of the diet with inulin, namely, PE(P-34:1) [PE(P-16:0/18:1); PE(P-18:1/16:0)] and PE(P-18:0/22:6), the latter being the most abundant Pl species of the cortex (10.9% of the PlsEtn). Their abundance was decreased in the cortex of mice fed a diet supplemented with inulin compared to those fed a control diet.

The inulin-dependent effect on PlsEtn could have deleterious effects on the brain tissue since PE(P-18:0/22:6) constitutes a major reservoir of C22:6*n*-3 (DHA). Indeed, DHA and its derivatives are essential for the development and maintenance of brain structure and function [61]. Epidemiological studies also support a link between dietary intake of DHA and the development of brain diseases and disorders such as AD [61,62]. In addition, decreased amounts of PE(P-18:0/22:6) and PE(P-16:0/18:1) have been reported in the cerebrum of patients with AD [63]. Potential harmful effects of inulin have already been described. Dietary intake of inulin has been shown to aggravate colitis, exacerbate atherosclerosis, enhance hepatic inflammation and fibrosis, disturb hepatic and bile acid metabolism, and cause hepatocellular carcinoma in specific genetic contexts associated with dysbiosis [39,64,65,66,67]. In addition, we have recently shown that although inulin prevents some of the alterations in the hepatic fatty acid metabolism caused by chronic consumption of a high-fat diet (HFD), it also exacerbates others [34]. Indeed, inulin consumption prevented the HFD-induced increase in C16:1*n*-9 and C20:3*n*-6 as well as the HFD-induced modulation of expression and/or activity of enzymes involved in fatty acid biosynthesis (*Elovl2*, *Elovl5* and FADS2) in mouse liver. However, this dietary fiber also exacerbated the HFD-induced increase in the hepatic amount of C17:0.

To expand our understanding of the mechanisms underlying the inulin-dependent decrease in some Pl species in the cortex, we explored several hypotheses. Since the liver has been proposed as the primary site of Pl biosynthesis, we investigated whether alterations of Pls in the cortex could have a hepatic origin. To this end, we evaluated the DMA content of the liver as well as the expression level of genes encoding key enzymes involved in the initial three steps of Pl biosynthesis (*Far1*, *Gnpat* and *Agps*) and compared them between mice fed a control diet and those fed an inulin-supplemented diet. Only DMA 16:0 was detected in the liver of mice at very low levels, which is consistent with previous studies [8]. No effect of inulin was observed, neither on the DMA content nor on the expression levels of *Far1*, *Gnpat* and *Agps* in the liver. This is in line with previous results showing no modification in the expression levels and activities of enzymes involved in the biosynthesis of fatty acids following inulin supplementation [34]. To go further in the exploration of a hepatic origin for the changes we observed in the Pl content of the cortex, we analyzed the Pls in the plasma. Indeed, we previously showed that the inulin supplementation, as we provided in the diet of this study, induced changes in the composition of the gut microbiota [34], and changes in the composition of the gut microbiota following inulin consumption have been associated with serum Pl levels [68]. No modification of the total DMA content was observed in the plasma of mice fed a diet supplemented with inulin. However, intra-class modifications were observed: the relative abundance of DMA 16:0 was decreased and counterbalanced by an increase in DMA 18:0. Whereas this result might account for the decrease in PE(P-34:1) [PE(P-16:0/18:1); PE(P-18:1/16:0] in the cortex, it does not explain that in PE(P-18:0/22:6). Altogether, these data suggest that it is unlikely that the Pl changes observed in the cortex have an extra-brain or hepatic origin.

Another hypothesis that could explain the decreased abundance of PlsEtn PE(P-18:0/22:6) and PE(P-34:1) [(PE(P-16:0/18:1); PE(P-18:1/16:0)] in the cortex of inulin-fed mice is a modulation of the endogenous biosynthesis of Pls. However, no modification in the expression levels of *Far-1*, *Gnpat* and *Agps* was observed in the cortex of mice fed an inulin-supplemented diet.

The bioavailability of the fatty acids in the cortex required for their biosynthesis was also analyzed. We observed no modification in the abundance of C16:0, C18:0, C18:1*n*-7, C18:1*n*-9 or C22:6*n*-3 in the cortex of mice fed an inulin-supplemented diet. However, the level of docosapentaenoic acid (DPA) from the *n*-3 series (C22:5*n*-3), which is an intermediate between eicosapentaeinoic acid (EPA, C20:5*n*-3) and DHA (C22:6*n*-3), was decreased. Finally, as the decrease in the abundance of some PlsEtn could also be the consequence of their hydrolysis, the expression levels of enzymes involved in Pl cleavage/degradation and the level of lyso species were evaluated. No modification of the expression level of *Pla2g6* and *Tmem86b* genes, encoding phospholipase A(2) and lysoplasmalogenase, respectively, was observed in mice fed an inulin-supplemented diet. Another cause of Pl degradation could be an attack on the vinyl–ether bond by oxidative stress-related molecules [47,48]. To test this hypothesis, the level of oxidative stress as well as the amount of oxidized derivatives of Pls should be evaluated. However, our results showed that dietary intake of inulin did not modify the expression level of a set of genes involved in oxidative stress-related mechanisms (*Cox-2*, *Cat*, *Gpx1*, *Sod1*, *Nos2* and *Sqstm1*) in the cortex. In addition, no lyso form of PlsEtn was detected in the cortex of mice fed a control diet or an inulin-supplemented diet and no modification of the ratio of LPEs/PEs was observed in the cortex of mice exposed to inulin. Taken together, these data suggest that the dietary intake of inulin does not enhance glycerophospholipid hydrolysis. However, as intermediate products arising from Pl degradation may only have a short-lived existence, further experiments such as assessment of phospholipase A(2) and lysoplasmalogenase activities are needed to rule out the existence of an impact of inulin consumption on PlsEtn degradation.

Finally, despite the use of compositionally controlled diets, we cannot exclude that the amount of fiber consumed by mice fed an inulin-supplemented diet was different to that of the control mice that received a cellulose-containing diet. Indeed, we have previously reported that inulin supplementation can slightly decrease food consumption, very likely linked to the energy provided by fermentable fiber compared to non-fermentable fiber [32]. More importantly, the dose of inulin used in the current study is relatively high and cannot be transposed to human nutrition. Hence, future studies appear warranted to investigate the effect of lower doses of inulin on cortex Pls.

## 5. Conclusions

In this study, we showed that dietary supplementation with inulin do not modify the global amount of Pls in the cortex of mice but affects its content at the species level. In particular, dietary intake of this prebiotic induces a decrease in the abundance of the most widely represented PlsEtn species, PE(P-18:0/22:6), which represents a major reservoir of DHA, a fatty acid essential for brain development and function. This study joins others that suggest inulin may have deleterious effects. The consequences of these alterations on the physiology and the functioning of the brain, as well as the molecular mechanisms that link inulin/gut microbiota and Pl levels in the brain, remain to be elucidated.

## Figures and Tables

**Figure 1 nutrients-14-03097-f001:**
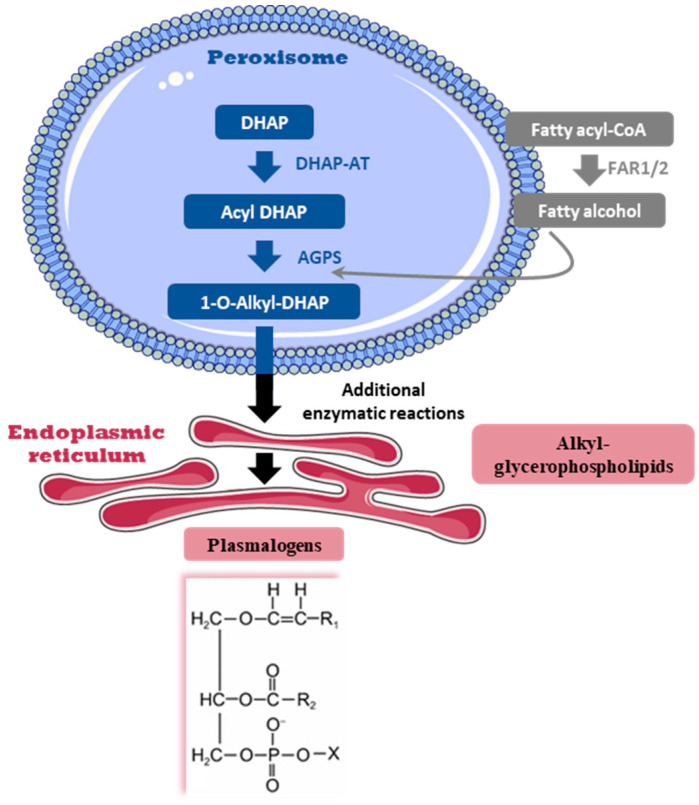
Schematic representation of plasmalogen biosynthesis. The biosynthesis of plasmalogens (Pls) is initiated in the peroxisome with three critical steps catalyzed by the enzymes FAR1 (fatty acyl-CoA reductase 1), DHAP-AT (dihydroxyacetone phosphate acyltransferase) and alkyl-DHAP synthase (alkylglycerone-phosphate synthase, AGPS). The biosynthesis of Pls is then continued in the endoplasmic reticulum by additional enzymatic reactions leading to the synthesis of alkyl-glycerophospholipid intermediates. The chemical structure of Pls is presented here. R1 denotes the carbon chain at the *sn*-1 position, and R2 at the *sn*-2 position. The polar head group, denoted by X, is most commonly choline or ethanolamine. acyl-CoA, acyl coenzyme A.

**Figure 2 nutrients-14-03097-f002:**
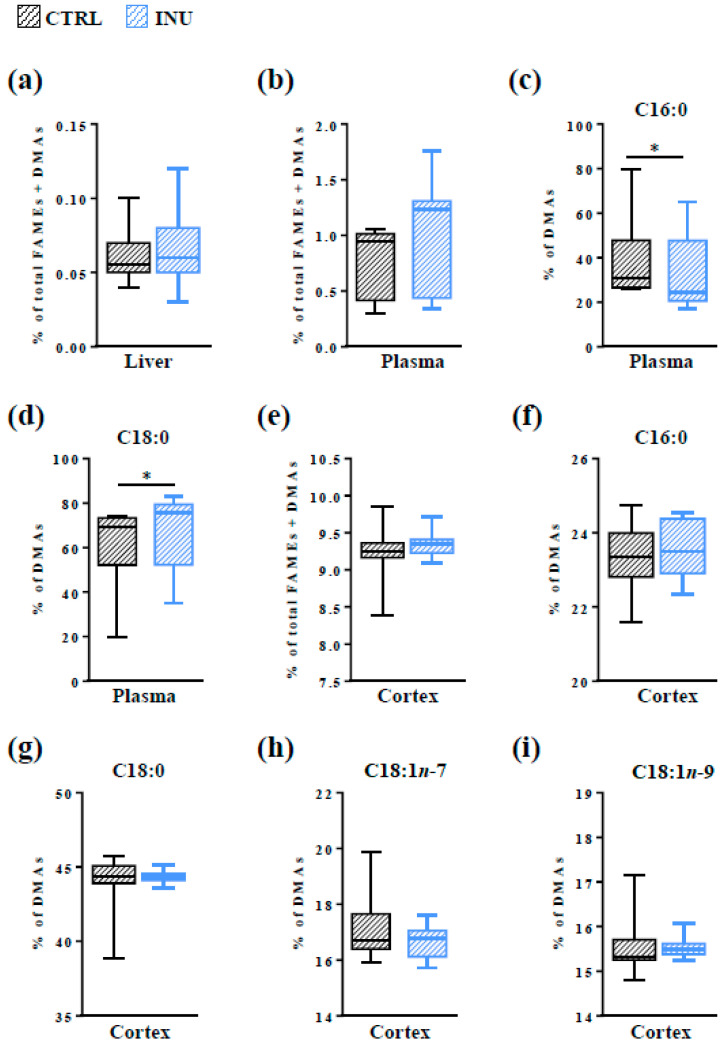
Evaluation of plasmalogen (Pl) content in the liver, plasma and cortex by GC-FID. The results represent the quantification of dimethylacetals (DMAs, derivatives of aldehyde aliphatic groups from the *sn*-1 position of Pls) by GC-FID. (**a**,**b**,**e**) Results are expressed as percentages of total DMAs relative to total fatty acid methyl esters (FAMEs) + total DMAs, defined as 100%, in the liver (**a**), in the plasma (**b**) and in the cortex (**e**). (**c**,**d**,**f**–**i**) percentages of (**c**,**f**) DMA 16:0, (**d**,**g**) DMA 18:0, (**h**) DMA 18:1*n*-7, and (**i**) DMA 18:1*n*-9 relative to total DMAs (defined as 100%), in the plasma (**c**,**d**) and in the cortex (**f**–**i**). CTRL: mice fed a control diet. INU: mice fed a diet supplemented with inulin. Data are presented in box and whisker plot format (median; min. to max.). Mann–Whitney test for comparison of lipid abundance between CTRL and INU mice, * *p* < 0.05. GC-FID, gas chromatography with flame-ionization detection.

**Figure 3 nutrients-14-03097-f003:**
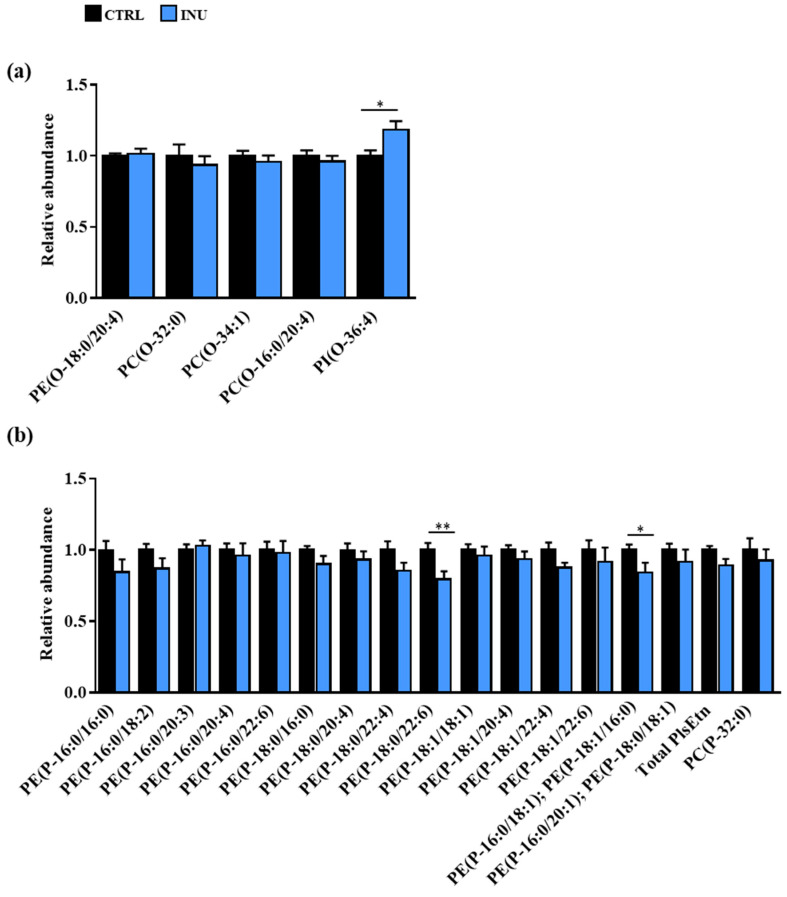
Changes in the abundance of AKG and Pl species in the cortex of mice fed a diet supplemented with inulin. (**a**) AKGs and (**b**) Pls. Results are expressed as fold change in the cortex of INU mice relative to the mean level observed in the cortex of CTRL mice, defined as 1.0. All data are presented as mean ± SEM. Mann–Whitney test for comparison of each AKG or Pl species abundance between CTRL and INU mice, ** p* < 0.05, *** p* < 0.01. AKG, alkyl-glycerophospholipid; PI, Plasmalogen; SEM, standard error of the mean.

**Figure 4 nutrients-14-03097-f004:**
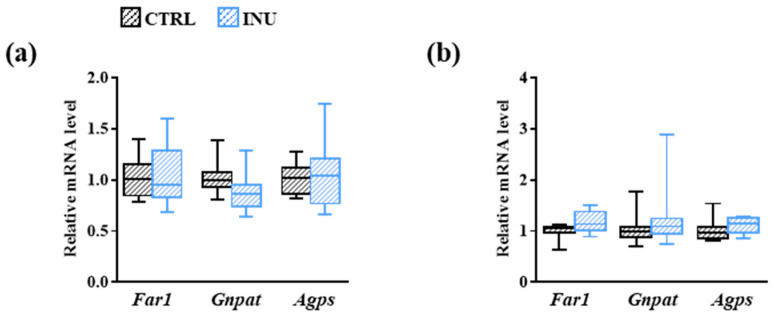
Effect of inulin on the expression of genes encoding enzymes involved in the biosynthesis of plasmalogens. Liver (**a**) and cortex (**b**) expression of genes encoding fatty acyl-CoA reductase 1 (*Far1*), DHAP-AT/DAP-AT (*Gnpat*), and alkyl-DHAP synthase (*Agps*) in mice fed a control diet or a diet supplemented with inulin. The levels of mRNA were normalized to *Hprt* mRNA level for calculation of the relative levels of transcripts. mRNA levels are illustrated as fold change. Data are presented in box and whisker plot format (median, min. to max.). Mann–Whitney test for comparison of the level of each mRNA between CTRL and INU mice.

**Figure 5 nutrients-14-03097-f005:**
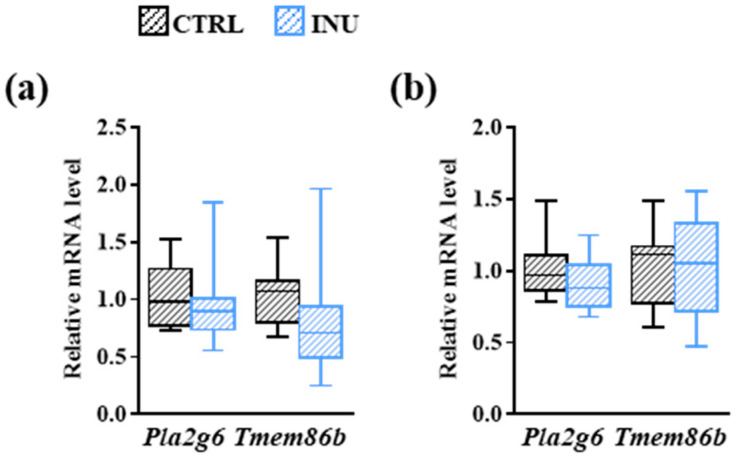
Effect of inulin on the expression of gene-encoding enzymes involved in the degradation of plasmalogens. Liver (**a**) and cortex (**b**) expression of genes encoding phospholipase A(2) (*Pla2g6*) and lysoplasmalogenase (*Tmem86b*) in mice fed a control diet or a diet supplemented with inulin. The levels of mRNA were normalized to *Hprt* mRNA level for calculation of the relative levels of transcripts. mRNA levels are illustrated as fold change. Data are presented in box and whisker plot format (median, min. to max.). Mann-Whitney test for comparison of the level of each mRNA between CTRL and INU mice.

**Figure 6 nutrients-14-03097-f006:**
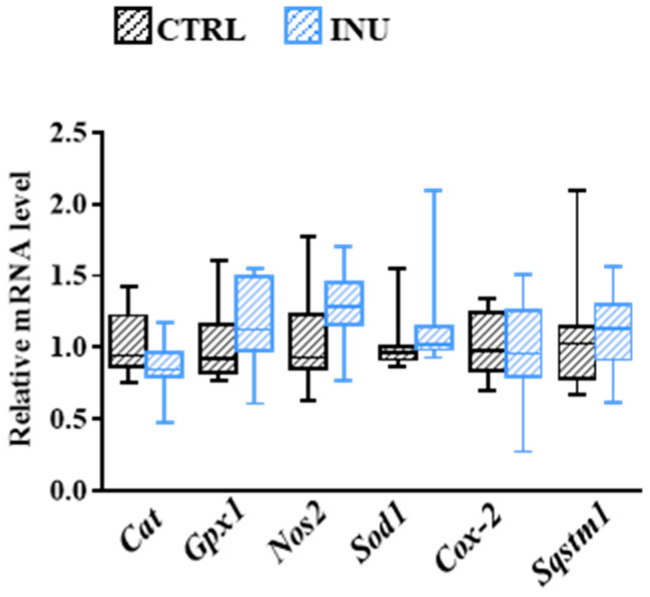
Effect of inulin on the cortex expression of gene-encoding proteins involved in oxidative stress-related mechanisms. *Cat* encodes catalase, *Gpx1* encodes glutathione peroxidase 1, *Nos2* encodes inducible nitric oxide (NO) synthase, *Sod1* encodes superoxide dismutase (Cu-Zn), *Cox-2* encodes cyclooxygenase-2, and *Sqstm1* encodes sequestosome-1 (ubiquitin-binding protein p62). The levels of mRNA were normalized to *Hprt* mRNA level for calculation of the relative levels of transcripts. mRNA levels are illustrated as fold change. Data are presented in box and whisker plot format (median; min. to max.). The Mann–Whitney test was used for comparison of the level of each mRNA between CTRL and INU mice.

**Table 1 nutrients-14-03097-t001:** Primer sequences.

Genes (ID)	Sense (5′-3′)	Antisense (5′-3′)
*Hprt* (15452)	CAGTCCCAGCGTCGTGATTA	TGGCCTCCCATCTCCTTCAT
*Far1* (67420)	GCTCGGAAGCATCTCAACAAG	GTGCTGGATGCTCGGAAGTAT
*Gnpat* (14712)	TCACCGCAGCTACATTGACT	GCAGCTCACTGACCACTCTC
*Agps* (228061)	GTGCAGGGTGACACAGACTT	CCATGGTGATGTGACAGGCT
*Pla2g6* (53357)	AAAGTCCCCTCAAGTGCCTG	ACAGTCCACGACCATCTTGC
*Tmem86b* (68255)	TGGGGTGCTGTGCTCTTTAC	CACTAGGCGGGCAAAAGGTA
*Cat* (12359)	CAACAGCTTCAGCGCACCAG	GGCCGGCAATGTTCTCACAC
*Gpx1* (14775)	GGAATGCCTTGCCAACACCC	GTCGATGGTACGAAAGCGGC
*Nos2* (18126)	AGAGCCACAGTCCTCTTTGC	ACCACCAGCAGTAGTTGCTC
*Sod1* (20655)	GATGAAAGCGGTGTGCGTGC	TGGACGTGGAACCCATGCTG
*Cox-2* (19225)	TTGCATTCTTTGCCCAGCAC	TTAAGTCCACTCCATGGCCC
*Sqstm1* (18412)	TAAAAGCTGGGCTCTCGGCG	CGTGAACGACGCCATAACCG

**Table 2 nutrients-14-03097-t002:** Relative amounts of alkyl-glycerophospholipid and plasmalogen species in the different classes of glycerophospholipids measured in mouse cerebral cortex.

Glycerophospholipids	Relative Abundance (%)
**Ethanolamine glycerophospholipids**	
***Alkyl-glycerophospholipids***	
PE(O-18:0/20:4)	0.619 ± 0.011
***Plasmalogens***	
PE(P-16:0/16:0)	0.178 ± 0.011
PE(P-16:0/18:2)	0.233 ± 0.010
PE(P-16:0/20:3)	0.365 ± 0.014
PE(P-16:0/20:4)	1.546 ± 0.071
PE(P-16:0/22:6)	5.189 ± 0.299
PE(P-18:0/16:0)	0.715 ± 0.021
PE(P-18:0/20:4)	5.001 ± 0.230
PE(P-18:0/22:4)	2.084 ± 0.126
PE(P-18:0/22:6)	10.857 ± 0.530
PE(P-18:1/18:1)	4.206 ± 0.172
PE(P-18:1/20:4)	3.240 ± 0.109
PE(P-18:1/22:4)	2.250 ± 0.117
PE(P-18:1/22:6)	2.757 ± 0.187
PE(P-16:0/18:1); PE(P-18:1/16:0) *	3.855 ± 0.147
PE(P-16:0/20:1); PE(P-18:0/18:1) *	3.681 ± 0.164
Total PlsEtn	46.155 ± 1.303
**Choline glycerophospholipids**	
***Alkyl-glycerophospholipids***	
PC(O-32:0)	0.134 ± 0.011
PC(O-34:1)	0.361 ± 0.013
PC(O-16:0/20:4)	0.108 ± 0.004
***Plasmalogens***	
PC(P-32:0)	0.125 ± 0.010
**Inositol glycerophospholipids**	
***Alkyl-glycerophospholipids***	
PI(O-16:0/20:4)	0.109 ± 0.004

For each glycerophospholipid class, results are expressed as abundance (in percentage) of each species relative to that of total species, defined as 100%. Data are expressed as mean ± SEM. * based on ion precursor fragmentation information given for both molecular ions according to their fatty acids moiety position. SEM, standard error of the mean.

**Table 3 nutrients-14-03097-t003:** Relative abundance of ester-linked fatty acids in the cortex.

Fatty Acids	CTRL	INU
**Saturated fatty acids (SFAs)**		
C14:0	0.143 ± 0.004	0.136 ± 0.004
C15:0 ***	0.044 ± 0.001	0.059 ± 0.003
C16:0	22.033 ± 0.153	22.014 ± 0.167
C17:0 ****	0.146 ± 0.002	0.181 ± 0.006
C18:0	21.454 ± 0.081	21.471 ± 0.059
C20:0	0.273 ± 0.009	0.274 ± 0.006
C22:0	0.183 ± 0.006	0.181 ± 0.009
C24:0	0.207 ± 0.008	0.219 ± 0.017
Total	44.482 ± 0.170	44.535 ± 0.159
**Monounsaturated fatty acids (MUFAs)**		
C16:1*n*-7 **	0.735 ± 0.016	0.671 ± 0.022
C18:1*n*-7	4.102 ± 0.042	4.038 ± 0.033
C20:1*n*-7	0.384 ± 0.013	0.364 ± 0.011
C16:1*n*-9	0.173 ± 0.002	0.167 ± 0.002
C18:1*n*-9	17.532 ± 0.170	17.534 ± 0.119
C20:1*n*-9	1.548 ± 0.057	1.574 ± 0.049
C22:1*n*-9	0.152 ± 0.005	0.148 ± 0.005
C24:1*n*-9	0.450 ± 0.019	0.454 ± 0.030
Total *n*-7 MUFAs (*p* = 0.0572)	5.221 ± 0.066	5.073 ± 0.056
Total *n*-9 MUFAs	19.854 ± 0.239	19.878 ± 0.188
Total MUFAs	25.076 ± 0.271	24.950 ± 0.216
**Polyunsaturated fatty acids (PUFAs)**		
C20:5*n*-3	0.062 ± 0.002	0.061 ± 0.002
C22:5*n*-3 *	0.168 ± 0.002	0.157 ± 0.003
C22:6*n*-3	15.360 ± 0.172	15.328 ± 0.111
C18:2*n*-6 **	0.660 ± 0.022	0.570 ± 0.014
C20:2*n*-6	0.094 ± 0.005	0.088 ± 0.005
C20:3*n*-6 ****	0.463 ± 0.005	0.411 ± 0.010
C20:4*n*-6	10.598 ± 0.072	10.768 ± 0.074
C22:4*n*-6	2.551 ± 0.024	2.590 ± 0.020
C22:5*n*-6 *	0.303 ± 0.006	0.351 ± 0.023
C20:3*n*-9 **	0.127 ± 0.003	0.141 ± 0.004
Total *n*-3 PUFAs	15.590 ± 0.171	15.546 ± 0.111
Total *n*-6 PUFAs	14.668 ± 0.083	14.778 ± 0.091
Total PUFAs	30.386 ± 0.249	30.466 ± 0.163
*n*-6 PUFAs/*n*-3 PUFAs	0.942 ± 0.006	0.951 ± 0.008

The percentage of each fatty acid methyl ester (FAME) relative to that of total FAMEs (100%) was determined. Data are expressed as mean ± SEM. Mann-Whitney test for comparison of the abundance of each fatty acid between control group (CTRL) and inulin group (INU) mice, * *p* < 0.05, ** *p* < 0.01, *** *p* < 0.001, and **** *p* < 0.0001. SEM, standard error of the mean.

**Table 4 nutrients-14-03097-t004:** Relative abundance of lyso-phosphatidylethanolamine species in the cortex.

	CTRL	INU
^a^ LPE 14:0 *	0.274 ± 0.053	0.259 ± 0.131
LPE 16:0 (*p* = 0.0576)	5.485 ± 0.553	4.438 ± 0.462
LPE 18:0	7.184 ± 1.712	5.552 ± 1.782
LPE 20:0	0.653 ± 0.044	0.522 ± 0.068
LPE 22:0	0.034 ± 0.004	0.032 ± 0.005
LPE 14:1	0.016 ± 0.004	0.026 ± 0.010
LPE 16:1	0.712 ± 0.051	0.652 ± 0.043
LPE 18:1	21.223 ± 0.722	21.005 ± 0.914
LPE 19:1	0.113 ± 0.009	0.121 ± 0.012
LPE 20:1	8.779 ± 0.396	8.129 ± 0.732
LPE 22:1	0.760 ± 0.046	0.695 ± 0.077
LPE 18:2	2.624 ± 0.502	2.463 ± 0.440
LPE 20:2	0.500 ± 0.049	0.513 ± 0.037
LPE 22:2	0.134 ± 0.011	0.127 ± 0.014
LPE 18:3	0.116 ± 0.013	0.136 ± 0.025
LPE 20:3	0.907 ± 0.078	0.938 ± 0.052
LPE 20:4	13.377 ± 0.520	14.498 ± 0.589
LPE 22:4	7.068 ± 0.830	8.058 ± 0.715
LPE 20:5	0.060 ± 0.008	0.059 ± 0.007
LPE 22:5	0.626 ± 0.045	0.750 ± 0.063
LPE 22:6	29.357 ± 1.145	31.032 ± 1.146
^b^ LPEs/PEs	0.807 ± 0.273	1.096 ± 0.487

^a^ The percentage of each lyso-phosphatidylethanolamine (LPE) species relative to that of total LPEs (100%) was determined. ^b^ Ratio of total LPEs/total ethanolamine glycerophospholipids (PEs, non-plasmalogens). Total LPEs and total PEs were calculated as the sum of the area under the peak of each species corrected with that of the internal control (PE(28:0)). Data are expressed as mean ± SEM. * *p* < 0.05. Mann–Whitney test for comparison of the abundance of LPE species or LPEs/PEs between CTRL and INU mice. SEM, standard error of the mean.

## Data Availability

The data presented in this study are available on request from the corresponding author.

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
