# Peer review of "Dietary Inulin Supplementation Affects Specific Plasmalogen Species in the Brain"

_nutrients, 2022, doi:10.3390/nu14153097_

Round 1

Reviewer 1 Report

In this manuscript, the authors have evaluated the effect of inulin supplementation on brain plasmalogens. The results are pretty mediocre. Besides, in the discussion, they mentioned the potentially harmful effects of inulin and that the dietary intake of inulin has been shown to aggravate colitis, exacerbate atherosclerosis, enhance hepatic inflammation and fibrosis, disturb hepatic and bile acid metabolism, and cause hepatocellular carcinoma in specific genetic contexts associated with dysbiosis [38,60-63]. 

In addition, they have recently shown that although inulin prevents some of the alterations in the hepatic fatty acid metabolism caused by chronic consumption of a high-fat diet, it also exacerbates others.

Therefore, in my opinion, a modest setting of plasmalogens don't justify the use of inulin to improve cerebral phospholipids. Also, the treatment had no global modification effect in the Pl amounts.   

However, if anything, the authors could study the inulin effects on the morphology of the cerebral cortex and neuronal survival. 

Author Response

REVIEWER 1

In this manuscript, the authors have evaluated the effect of inulin supplementation on brain plasmalogens. The results are pretty mediocre. Besides, in the discussion, they mentioned the potentially harmful effects of inulin and that the dietary intake of inulin has been shown to aggravate colitis, exacerbate atherosclerosis, enhance hepatic inflammation and fibrosis, disturb hepatic and bile acid metabolism, and cause hepatocellular carcinoma in specific genetic contexts associated with dysbiosis [38,60-63]. 

In addition, they have recently shown that although inulin prevents some of the alterations in the hepatic fatty acid metabolism caused by chronic consumption of a high-fat diet, it also exacerbates others.

Therefore, in my opinion, a modest setting of plasmalogens don't justify the use of inulin to improve cerebral phospholipids. Also, the treatment had no global modification effect in the Pl amounts.   

However, if anything, the authors could study the inulin effects on the morphology of the cerebral cortex and neuronal survival. 

** We thank reviewer 1 for his/her time dedicated to evaluate our work. As raised by the reviewer and demonstrated in several studies, including some from authors of this paper, inulin supplementation could have detrimental effects on health in specific experimental models. Along similar lines, we demonstrated in this study that, although inulin did not modify the global amount of plasmalogens in the cortex, it modulated the abundance of specific plasmalogen species in this tissue. Particularly, the abundance of PE(P-18:0/22:6) was decreased by 20% in the cortex of mice exposed to inulin-supplemented diet compared to control mice, which we consider being neither insignificant, nor mediocre. Indeed, PE(P-18:0/22:6) is the most abundant ethanolamine plasmalogen species in the cortex and a reservoir of DHA (C22:6n-3), a fatty acid required for brain development and function. We agree with the reviewer that to go further in the determination of the effects of inulin, « harmful versus beneficial », it will now be interesting to study the consequences of consuming this fiber chronically on brain structure and function. Careful modifications have been made in the conclusion paragraph in order to better transliterate these ideas (see p16, lines 534-539).

Reviewer 2 Report

In this work the authors showed the possible therapeutic effect of Inulin in brain functions. 

The work is well written, and the data are well represented. m

In-depth knowledge of the role of inulin could be crucial in the future in understanding brain damage prevention systems in humans. 

Author Response

REVIEWER 2

In this work the authors showed the possible therapeutic effect of Inulin in brain functions. 

The work is well written, and the data are well represented.

In-depth knowledge of the role of inulin could be crucial in the future in understanding brain damage prevention systems in humans. 

** We thank reviewer 2 for her/his positive opinion about our manuscript.

Reviewer 3 Report

General comments:

Several inherited diseases but also common sporadic neurodegenerative disorder and aging have been associated with reduced brain plasmalogen levels. Therapeutic studies with oral plasmalogen supplementation to restore the levels in the nervous system are ongoing – so far with limited success. Improved understanding of potential contributions from diet and gut microbiota in modulating the lipid profile of host tissues could be highly relevant. The authors of the current paper have a well-documented research interest in the interactions of diet and gut microbiota and their effects on  brain lipids.  

Here, Bizeau et al studied the effect of a diet containing 20% of the probiotic inulin on the plasmalogen content in the brain, liver and plasma of young adult mice. The results indicated no significant alterations of total plasmalogen levels in any of the tissues after 11 weeks of treatment. Also lyso-plasmalogen intermediates, arising during synthesis or degradation of plasmalogens, were detected at normal levels.  However, some of the major molecular species of ethanolamine plasmalogen were found to be downregulated in cerebral cortex. Analysis of gene expression in cortex and liver revealed no effect on the mRNA levels for key enzymes in plasmalogen biosynthesis or degradation. The study identifies differences in the lipid profile upon inulin ingestion but provides no further mechanistic insight into the interaction between inulin, gut bacteria and the brain plasmalogen profile.

In general, the research question and methodology are valid and no new experiments required. However, in the Discussion, line 454-459, the authors mention, as “data not shown”, that own “preliminary experiments” indicated no effect of dietary inulin on antioxidant-related gene expression. In most journals “data not shown” would not be an acceptable statement. Would it be possible to show these data - if not for the full gene set, at least for some of the listed genes - in the present paper?  The reviewer is not insisting on this point but it would add more weight to excluding oxidative stress-related mechanisms.

 Specific comments and questions

TITLE:  The title could be rephrased to deliver a more precise message: > that Inulin diet lowers specific plasmalogen species in the brain

INTRODUCTION 

1)  The current Fig. 2 would fit better (as Fig. 1) in the Introduction, for example in the 2nd paragraph, where structure and synthesis of plasmalogens are explained.

2)  To the references in line 58, add Brites et al (2011), which shows well that Plasmalogen precursor supplementation rescues the plasmalogen levels in the periphery but not in the CNS. There is no indication of efficient transfer across the BBB of these precursors in that study.  [Brites et al. (2011) Alkyl-Glycerol Rescues Plasmalogen Levels and Pathology of Ether-Phospholipid Deficient Mice. PLoS ONE 6(12): e28539. doi:10.1371].

3)  Line 63: Between “located in the outer” and “peroxisome membrane”, insert “surface of the”. Peroxisomes are lined by a single membrane, not double membranes like mitochondria, hence there is no outer membrane but an outer (sur)face.

4)  Lines 52-60 (concerns also Results and Discussion): The concept that the liver provides Pls for the other tissues is controversial. Please, rephrase throughout the manuscript (e.g., also at the beginning of Results and in the Discussion, line 418-419) to avoid stating this as a fact. For example, instead of : “Since the liver is the primary site of Pl synthesis” modify to “ Since liver has been proposed as the primary … “ or similar. 

Local, endogenous synthesis is a more likely main source, in particular for the brain. The levels of Pls are very high in the brain but very low in the liver; also the genes for the synthesizing enzymes are highly expressed in the brain, while the mRNAs are of low abundance in the liver. Presumably, also the authors’ own experiments confirm the large difference in expression between liver and brain tissue (see comment the display of the RT-qPCR results below, in the Results section).

 MATERIALS AND METHODS

1)  Section 2.1. Add a statement for the plasma collection.

3)  Section 2.5. Statistical Analysis:  For optimal display of the data in Figs 4 and 5, column graphs with error bars are suboptimal. Scatter/dot plots with mean/median and error bars (or at least Box-and-whisker plots) would be preferable to enable better interpretation of the data.

Each figure legend should have an explicit statement for the statistical test used. Was post-hoc correction for multiple comparisons performed?

RESULTS

1)  Fig. 2. The protein nomenclature for GNPAT/DHAP-AT and AGPS/Acyl DHAP synthase (ADHAPS?) is used inconsistently in the figure and the legend/text. Indeed, these complicated names are always a bit problematic, especially when the official gene symbols don’t match the historical protein symbols. 

2)  Table 2, Title: Specify the tissue analyzed (murine cerebral cortex).

3) Figs. 4 and 5:  For the RT-qPCR results, only fold-change of relative mRNA levels is shown. An indication of the relative expression (ratio to Hprt) would provide some valuable information about the abundance in brain versus liver (see Introduction, comment 4). This could easily be added as a regular or supplementary Table.

DISCUSSION

Line 414-416: The statement here is rather vague. Please, specify the alterations in fatty acid metabolism that were observed to be prevented and exacerbated by inulin in Ref. #33. Although, in a context of high fat-diet, the specific information may still be of interest here. Some fatty acids could, for example, activate or polarize microglia, with subsequent changes in the lipid profile of the brain.

Line 468-473: Concerning the last paragraph of the Discussion: Would it not be routine documentation in a feeding experiment to monitor feed consumption (at least per cage to obtain an average per mouse) and body weight. Was this not the case here?

Finally, a general basic question to the feeding regime and its potential translation to human dietary supplementation: The mice received inulin as a major component (20% w/w) of their diet; how much would be realistic as supplement in human diets?

 Minor comments, language and proofreading: 

Line 58:  Delete “others”. The brain would be the “central” organ. 

Line 145; In “fatty methyl esters (FAME)”, “acid” is missing.

Line 224: modified > modify

Line 309: gene > genes

Line 310-312 Fig 4 (legend) and 343-344:  Gene symbols should be in italics.

Line 445: For clarity, change “their biosynthesis” to  the biosynthesis of Pls.

Line 451: Exchange the period with a comma, to connect the two “half-sentences”.

Author Response

REVIEWER 3

General comments:

Several inherited diseases but also common sporadic neurodegenerative disorder and aging have been associated with reduced brain plasmalogen levels. Therapeutic studies with oral plasmalogen supplementation to restore the levels in the nervous system are ongoing – so far with limited success. Improved understanding of potential contributions from diet and gut microbiota in modulating the lipid profile of host tissues could be highly relevant. The authors of the current paper have a well-documented research interest in the interactions of diet and gut microbiota and their effects on  brain lipids. 

Here, Bizeau et al studied the effect of a diet containing 20% of the probiotic inulin on the plasmalogen content in the brain, liver and plasma of young adult mice. The results indicated no significant alterations of total plasmalogen levels in any of the tissues after 11 weeks of treatment. Also lyso-plasmalogen intermediates, arising during synthesis or degradation of plasmalogens, were detected at normal levels.  However, some of the major molecular species of ethanolamine plasmalogen were found to be downregulated in cerebral cortex. Analysis of gene expression in cortex and liver revealed no effect on the mRNA levels for key enzymes in plasmalogen biosynthesis or degradation. The study identifies differences in the lipid profile upon inulin ingestion but provides no further mechanistic insight into the interaction between inulin, gut bacteria and the brain plasmalogen profile.

In general, the research question and methodology are valid and no new experiments required.

** We thank reviewer 3 for her/his positive opinion about our manuscript, her/his careful reading and her/his relevant and constructive comments.

However, in the Discussion, line 454-459, the authors mention, as “data not shown”, that own “preliminary experiments” indicated no effect of dietary inulin on antioxidant-related gene expression. In most journals “data not shown” would not be an acceptable statement. Would it be possible to show these data - if not for the full gene set, at least for some of the listed genes - in the present paper?  The reviewer is not insisting on this point but it would add more weight to excluding oxidative stress-related mechanisms.

** As suggested by the reviewer, we have added a new Figure (Figure 6, p14) in order to present the expression level in the cortex of INU mice compared to CTRL mice of several genes involved in oxidative stress-related mechanisms. A new paragraph has been added in the Results section (see p13, lines 399-409) and the mention « data not shown » has been removed in the Discussion (see p16, line 517). Primers used are listed in Table 1, p6.

 Specific comments and questions

TITLE:  The title could be rephrased to deliver a more precise message: > that Inulin diet lowers specific plasmalogen species in the brain

** The title has been edited accordingly to: « Dietary inulin supplementation affects specific plasmalogen species in the brain »

INTRODUCTION 

1)  The current Fig. 2 would fit better (as Fig. 1) in the Introduction, for example in the 2nd paragraph, where structure and synthesis of plasmalogens are explained.

** We thank the reviewer for this judicious suggestion. Figures 1 and 2 have been inverted and Figure 1 (Schematic representation of plasmalogen biosynthesis) is now cited in the Introduction section (see p2, line 68).

2)  To the references in line 58, add Brites et al (2011), which shows well that Plasmalogen precursor supplementation rescues the plasmalogen levels in the periphery but not in the CNS. There is no indication of efficient transfer across the BBB of these precursors in that study.  [Brites et al. (2011) Alkyl-Glycerol Rescues Plasmalogen Levels and Pathology of Ether-Phospholipid Deficient Mice. PLoS ONE 6(12): e28539. doi:10.1371].

** This important reference has been added.

3)  Line 63: Between “located in the outer” and “peroxisome membrane”, insert “surface of the”. Peroxisomes are lined by a single membrane, not double membranes like mitochondria, hence there is no outer membrane but an outer (sur)face.

** We thank the reviewer for bringing this to our attention. This clarification has been made (see p2 line 70).

4)  Lines 52-60 (concerns also Results and Discussion): The concept that the liver provides Pls for the other tissues is controversial. Please, rephrase throughout the manuscript (e.g., also at the beginning of Results and in the Discussion, line 418-419) to avoid stating this as a fact. For example, instead of : “Since the liver is the primary site of Pl synthesis” modify to “ Since liver has been proposed as the primary … “ or similar. 

Local, endogenous synthesis is a more likely main source, in particular for the brain. The levels of Pls are very high in the brain but very low in the liver; also the genes for the synthesizing enzymes are highly expressed in the brain, while the mRNAs are of low abundance in the liver. Presumably, also the authors’ own experiments confirm the large difference in expression between liver and brain tissue (see comment the display of the RT-qPCR results below, in the Results section).

** We agree with the reviewer and we have accordingly rephrased the sentences (see p2 lines 56-67 in the Introduction section, p6 line 232 in the Results section and p15 line 476 in the Discussion section).

 MATERIALS AND METHODS

1)  Section 2.1. Add a statement for the plasma collection.

** This information has been added in our revised manuscript (see p4 lines 160-161).

2)  Section 2.5. Statistical Analysis:  For optimal display of the data in Figs 4 and 5, column graphs with error bars are suboptimal. Scatter/dot plots with mean/median and error bars (or at least Box-and-whisker plots) would be preferable to enable better interpretation of the data. Each figure legend should have an explicit statement for the statistical test used. Was post-hoc correction for multiple comparisons performed?

** The column graphs of Figures 4 (p10) and 5 (p12) have been edited. The data are now displayed in a box-and-whisker plot format. A more detailed description of the test applied is described now in each Figure legend.

RESULTS

1)  Fig. 2. The protein nomenclature for GNPAT/DHAP-AT and AGPS/Acyl DHAP synthase (ADHAPS?) is used inconsistently in the figure and the legend/text. Indeed, these complicated names are always a bit problematic, especially when the official gene symbols don’t match the historical protein symbols.

** We agree with this reviewer that there is not always a logic between the name of a protein and that of the corresponding gene. We have chosen to follow the recommendations (recommended names/short names for genes and proteins) from the Universal Protein Resource (Uniprot). We have harmonized the way proteins are cited in the text to facilitate the understanding.

2)  Table 2, Title: Specify the tissue analyzed (murine cerebral cortex).

** The title of Table 2 is now: “Relative amounts of alkyl-glycerophospholipid and plasmalogen species in the different classes of glycerophospholipids measured in mouse cerebral cortex.” We thank the reviewer for bringing this to our attention.

3) Figs. 4 and 5:  For the RT-qPCR results, only fold-change of relative mRNA levels is shown. An indication of the relative expression (ratio to Hprt) would provide some valuable information about the abundance in brain versus liver (see Introduction, comment 4). This could easily be added as a regular or supplementary Table.

** As we did not perform absolute quantifications, we have chosen to present DeltaCt data in a new supplementary Figure A1 (p17). Although this presentation may have limitations, the results suggest lower expression levels of Far1, Gnpat and Agps in liver compared to cerebral cortex (see Results section p10 lines 319-323) and higher expression levels of Pla2g6 and Tmem86b in liver compared to cerebral cortex (see Results section p12 lines 365-366).

DISCUSSION

Line 414-416: The statement here is rather vague. Please, specify the alterations in fatty acid metabolism that were observed to be prevented and exacerbated by inulin in Ref. #33. Although, in a context of high fat-diet, the specific information may still be of interest here. Some fatty acids could, for example, activate or polarize microglia, with subsequent changes in the lipid profile of the brain.

** We have given more detailed information on changes in the fatty acid metabolism that occurred in liver when mice were exposed to an inulin-supplemented HFD compared to a HFD (see p15 lines 468-472)

Line 468-473: Concerning the last paragraph of the Discussion: Would it not be routine documentation in a feeding experiment to monitor feed consumption (at least per cage to obtain an average per mouse) and body weight. Was this not the case here?

** We thank the reviewer for this important comment. We unfortunately did not monitor food consumption in the current work, since doing it precisely when using purified diet could be challenging. Indeed, mice manipulate and scramble food pellets, thus resulting in non-consumed diet accumulation in the litter. We nonetheless agree with the reviewer that this is an important aspect to monitor, and we will make sure to do it in our future studies.

Finally, a general basic question to the feeding regime and its potential translation to human dietary supplementation: The mice received inulin as a major component (20% w/w) of their diet; how much would be realistic as supplement in human diets?

** We thank the reviewer for this important comment, since the dose of inulin used in the current study is relatively high and cannot be transposed to the human diet. Some previous work from our co-authors has demonstrated that inulin detrimental impact on intestinal health is also observed at lower doses that are comparable to those found in human nutrition, but this aspect would need to be carefully investigated regarding brain plasmalogens in the future. This important limit to our current work has now been clearly mentioned in our revised discussion, page 16 lines 530-532.

 Minor comments, language and proofreading: 

Line 58:  Delete “others”. The brain would be the “central” organ.

** Done, thank you.

Line 145; In “fatty methyl esters (FAME)”, “acid” is missing.

** Done, thank you.

Line 224: modified > modify

** Done, thank you.

Line 309: gene > genes

** Done, thank you.

Line 310-312 Fig 4 (legend) and 343-344:  Gene symbols should be in italics.

** We checked that all the gene names were italicized.

Line 445: For clarity, change “their biosynthesis” to  the biosynthesis of Pls.

** Done, thank you.

Line 451: Exchange the period with a comma, to connect the two “half-sentences”.

** Done, thank you.

Round 2

Reviewer 1 Report

I have read the author's reply. In my opinion, their answers are not persuading. Besides, they didn't reply to my request:

However, if anything, the authors could study the inulin effects on the morphology of the cerebral cortex and neuronal survival. 

Author Response

Dear Editor in Chief,

We would like to thank again the reviewers for their comments and advices that constructively have helped for improving our manuscript (nutrients-1807840, by Bizeau et al). We thank reviewers 2 and 3 for their positive opinion regarding the changes we made in the revised version of the manuscript. However, we are surprised by the comments of reviewer 1, since we have endeavored a fair discussion with this reviewer.

Please find below our response.

Kind regards

Marie Agnès Bringer, PhD

REVIEWER 1. I have read the author's reply. In my opinion, their answers are not persuading. Besides, they didn't reply to my request: However, if anything, the authors could study the inulin effects on the morphology of the cerebral cortex and neuronal survival. 

** We understand that it would be interesting to go further in the evaluation of the inulin effect on the brain by analyzing for instance the morphology of the cerebral cortex as well as neuronal survival or other parameters, variables and/or biological processes. However, we would like to kindly remind reviewer 1 that this was not the aim of the present study. Indeed, as announced in the introduction section of the manuscript the objective was to investigate the impact of dietary intake of inulin on the plasmalogen content of the cerebral cortex and our study was focused on a very detailed analysis of these lipids that have already been extensively investigated in our laboratory.   

This manuscript is a resubmission of an earlier submission. The following is a list of the peer review reports and author responses from that submission.

Round 1

Reviewer 1 Report

The manuscript Dietary inulin supplementation affects brain plasmalogens by Jean-Baptiste Bizeau, Mayssa Albouery, Stéphane Grégoire, Bénédicte Buteau, Lucy Martine, Marine Crépin, Alain M Bron, Olivier Berdeaux, Niyazi Acar, Benoit Chassaing, Marie-Agnès Bringer

Is an interesting paper where authors explore the effects of inulin supplementation in mice over the content and composition of plasmalogens in the brain. However, some issues should be revised:

Major comments

  1. Authors’ results section begin with the effect of inulin over total Pls in the liver and brain, which is fine, but they should show the amount of diet and total energy consumption for each diet group, because it has been shown that sometimes diets are not completely pleasant for mice. This data is useful in terms of the amount of dietary fiber consumed for each group.
  2. In the section 3.3 authors demonstrate that Far1, Gnpat and Agps in the liver and brain cortex does not have a significant fold change, except for Agps in the cortex. Does the authors have evidence of the expression at the protein level (western-blot or immunostaining?)
  3. The Pls plasma levels would be of interest in order to show diet-derived differences
  4. Authors should consider the measurement of lysoplasmalogenase and phospholipase A2 enzymes due to its role on the Pls processing (PCR and/or western blot) in cortex and the liver tissues
  5. Does the authors have any information bout intestinal or fecal microbiota in order to establish pertinent correlations?
  6. In lane 442 authors wrote this paragraph “These findings support the role of the gut microbiota in the regulation of brain lipids” This reviewer strongly disagree with this. Authors do not show any evidence of their mice gut microbiota. In addition, the main goal was to establish the relationship between inulin consumption and Pls levels in brain and liver. In order to write the above sentences, authors should show the microbiota modification in terms of relative abundance or so.

Minor comments

  1. Sometimes Plasmalogens are abbreviated Pl and in lanes 164, 171, 183, 185 and others, appears as PL
  2. In the materials section gene expression, the authors should mention the amount of total RNA they used to perform the reverse transcription reaction as well as the relative quantitative method for amplicon estimation.
  3. Table 1, Gene ID for each gene should the mentioned
  4. The animal facility dark/light cycle conditions should be mentioned

Reviewer 2 Report

The manuscript entitled: Dietary inulin supplementation Aafects brain Plasmalogens by Bizeau et al. is quite interesting. The authors narrate in the introduction section the roles of Plasmalogens in the brain and liver, but they used only the cortex of their animals. In my opinion, it would have been more appropriate to use the whole brain rather than just the cortex. Besides, they completely omit the authorization of an ethical committee that approves the use and the treatment of the animals. It is important to highlight that without this kind of authorization is not possible to accept a manuscript for publication in any journal!